# Aqueous-Phase Formation of Two-Dimensional PbI_2_ Nanoplates for High-Performance Self-Powered Photodetectors

**DOI:** 10.3390/mi14101949

**Published:** 2023-10-19

**Authors:** Muhammad Imran Saleem, Perumalveeramalai Chandrasekar, Attia Batool, Jeong-Hwan Lee

**Affiliations:** 1Department of Materials Science and Engineering, Inha University, Incheon 22212, Republic of Korea; imransaleem@inha.ac.kr; 2School of Science, Minzu University of China, Beijing 100081, China; pv_chandrasekar@muc.edu.cn; 3Research Center for Materials Science, Beijing Institute of Technology, Beijing 100081, China; attiabatool53@gmail.com; 43D Convergence Center, Inha University, Incheon 22212, Republic of Korea

**Keywords:** green-chemistry principles, aqueous-phase synthesis, two-dimensional nanoplates, self-powered photodetector

## Abstract

The process of the aqueous synthesis of nanomaterials has gained considerable interest due to its ability to eliminate the need for complex organic solvents, which aligns with the principles of green chemistry. Fabricating nanostructures in aqueous solutions has gained recognition for its potential to develop ultrasensitive, low-energy, and ultrafast optoelectronic devices. This study focuses on synthesizing lead iodide (PbI_2_) nanoplates (NPs) using a water-based solution technique and fabricating a planar photodetector. The planar photodetectors (ITO/PbI_2_ NPs/Au) demonstrated a remarkable photosensitivity of 3.9 × 10^3^ and photoresponsivity of 0.51 mA/W at a wavelength of 405 nm. Further, we have carried-out analytical calculations for key performance parameters including open-circuit voltage (*V_oc_*), short-circuit current (*I_sc_*), on-off ratio, responsivity (*R*), and specific detectivity (*D**) at zero applied bias, while photodetector operating in self-powered mode. These values are as follows: *V_oc_* = 0.103 V*, I_sc_* = 1.93 × 10^−8^, on-off ratio = 10^3^, *R =* 4.0 mA/W, and *D** = 3.3 × 10^11^ Jones. Particularly, the asymmetrical output properties of ITO/PbI_2_ NPs/Au detector provided additional evidence of the effective creation of a Schottky contact. Therefore, the photodetector exhibited a photo-response even at 0 V bias (rise/decay time ~1 s), leading to the realization of self-powered photodetectors. Additionally, the device exhibited a rapid photo-response of 0.23/0.38 s (−5 V) in the visible range. This study expands the scope of aqueous-phase synthesis of PbI_2_ nanostructures, enabling the large-area fabrication of high-performance photodetectors.

## 1. Introduction

Recently, various high-performance nanomaterials have been synthesized by solution processes such as hot injection and low-temperature recrystallization techniques [1,2,3]. While these methods yield nanomaterials with remarkable photophysical properties, their reliance on highly toxic organic solvents poses significant environmental challenges [4,5,6]. In recent years, the RE100 Climate Group, whose primary objective is to accelerate the transition to widespread zero-carbon grids and achieve 100% renewable electricity, has influenced the participation of renowned electronics companies. By adopting environmentally friendly techniques, the collaborative efforts of the group aim to ensure a smooth and efficient transition towards widespread zero-carbon grids. For example, global companies aim to enhance their utilization of renewable energy, invest in innovative technologies, conduct investigations to develop energy-efficient products, augment water reuse practices, and explore the advancements in carbon capture technology. However, the current method employed to synthesize nanomaterials and device fabrication through various physical and chemical techniques could be more environmentally friendly by adopting the RE100 guidelines and green chemistry principles [7,8].

The aqueous synthesis of nanomaterials has emerged as an alternative to minimize environmental pollution and implement green chemistry guidelines. Moreover, the resulting nanomaterials are dispersed in aqueous media without phase transfer, facilitating their use in biological applications. Among the diverse range of nanomaterials, layered two-dimensional (2D) materials have garnered considerable attention owing to their potential to advance cutting-edge technologies such as electronics, optoelectronics, and sensing applications. Researchers have been actively exploring the unique properties exhibited by 2D materials, particularly in optoelectronics devices [9,10]. Using ultrabroad wavelength photodetectors, layered 2D materials have found successful applications in optical communication, sensing, and imaging. Particularly, the exceptional flexibility and adaptability of the layered 2D materials to various materials and substrates make them highly advantageous. Moreover, these materials exhibit a unique characteristic of being free from dangling bonds, which typically contributes to surface recombination and an increase in the dark current [11,12,13].

Despite their remarkable properties, the fabrication of 2D material structures is achieved by stacking exfoliated 2D flakes or 2D thin films grown by chemical vapor deposition (CVD), using a layer-by-layer transfer technique. However, this transfer method faces significant challenges such as potential interface contamination, limited scalability, and lack of precise control over stack configuration. Therefore, large-scale production of 2D material-based optoelectronic devices is imperative and relies heavily on improvements in processing techniques for advanced materials. Therefore, the development of a solution-processed approach is crucial, as it holds great promise for constructing novel optoelectronic devices with outstanding performance.

Recently, photodetectors are gaining interest, owing to their fundamental capability of converting incident light pulses into electrical signals [14,15,16,17,18,19]. Photodetectors find extensive applications in various fields such as infrared imaging, fire imaging, optical communications, biological sensing, and spectroscopy [20,21,22,23,24,25,26,27,28,29]. The remarkable photophysical characteristics and structural advantages exhibited by 2D layered materials have positioned them at the forefront of research for photodetection applications. Graphene has been a prominent choice in photodetection applications; however, the lack of a discernible bandgap restricts its applicability. Consequently, other 2D layered materials, including black phosphorus (BP), have emerged as promising alternatives. Nevertheless, the use of BP in photodetection is limited by stability concerns, particularly its susceptibility to irreversible oxidation when exposed to ambient conditions.

The pursuit of autonomously operating photodetectors, leveraging stable 2D layered materials within their design, holds considerable promise. They have the unique capability of harnessing energy from their surrounding environment and converting it into electrical pulses. This is an essential characteristic for practical applications owing to several key advantages, including low energy consumption, sustainability, and simplified design of driving circuits. Consequently, the development of self-powered photodetectors based on all-solution-processed 2D layered materials stands as a particularly crucial and innovative endeavor in the realm of photodetection technology. In this study, PbI_2_ nanoplates (NPs) with diverse morphologies, including hexagonal, truncated triangular, and tri-pyramidal NPs, were synthesized using an aqueous solution of PbI_2_. The synthesis involved simply dropping the precursor solution (prepared in DI water) onto a pre-cleaned indium tin oxide (ITO) substrate that was preheated to 90 °C to form thin films, enabling large-area fabrication as well. Subsequently, the temperature was elevated to 180 °C within 5 min to support nucleation and growth with distinct geometric configurations, including triangular, pyramidal, and hexagonal PbI_2_ NPs. The optoelectronic properties of PbI_2_ NPs were assessed by fabricating a planar photodetector ITO/PbI_2_ NP/Au device structure, which exhibited a high photosensitivity of 3.9 × 10^3^ and photoresponsivity of 0.51 mA/W (−2 V) at a wavelength of 405 nm. Moreover, the device demonstrated a rapid photo-response with a speed of 0.23/0.38 s within the visible range.

## 2. Materials and Methods

All the chemical reagents were of analytical grade and used as received without further purification. Lead (II) iodide (PbI_2_, 99.9985%) was purchased from Alfa Aesar (Haverhill, MA, USA).

### 2.1. Synthesis of 2D PbI_2_ Nanoplates

First, PbI_2_ (10 mg) was mixed in 10 mL of deionized water and magnetically stirred at 90 °C until the powder dissolved entirely. The solution became clear with negligible sedimentation. Subsequently, 50–60 µL of the clear supernatant solution was dropped onto an oxygen-plasma glass substrate in ambient conditions at 90 °C. Subsequently, the temperature was elevated to 180 °C within 5 min to support nucleation and growth with distinct geometric configurations, including triangular, pyramidal, and hexagonal PbI_2_ NPs.

### 2.2. Device Fabrication

Pre-patterned ITO glass substrates were each cleaned in order with acetone, isopropyl alcohol, and deionized water for 15 min in an ultrasonic bath. The substrates were treated with UV ozone to make the surface hydrophilic. Subsequently, the 2D PbI_2_ NPs were grown using the abovementioned drop-casting method. Finally, all devices were completed by thermally evaporating a 100 nm Au electrode. (The thermal evaporator located at the 3D Convergence Center of Inha University). The active area of the device was 4 mm^2^, defined by the intersection of ITO and Au stripes.

### 2.3. Characterizations

The X-ray diffraction (XRD) patterns of samples were obtained using the Rigaku Ultima IV X-ray diffractometer with Cu Kα radiation (λ = 1.5406 Å). Raman spectrum was measured by drop-casting PbI_2_ precursor solution on a pre-cleaned silicon substrate with a 532 nm wavelength laser. Scanning electron microscopy (SEM) images were acquired using the ZEISS Sigma 500 field-emission scanning electron microscope operated at an accelerating voltage of up to 30 kV. The optical UV-absorption spectra were measured with a spectrometer (Perkin Elmer, Lambda 750). The current-voltage (I-V) characteristics of the devices were measured using a Keithley 4200 SCS unit. A 405 nm laser diode was used as the light source for the photocurrent measurements, whereas the power of the incident radiation was tuned and measured using a power meter.

## 3. Results

The PbI_2_ NPs were synthesized using a low-temperature solution processing method (Figure 1a,b). The crystal structure of PbI_2_ is analogous to other transition metal dichalcogenides (such as MoS_2_, S-Mo-S), indicating that a layer of lead (Pb) atoms is covalently linked to two layers of iodide (I) atoms, forming the repeated hexagonal structure [30,31]. The interaction between adjacent layers with a spacing of 6.98 Å is governed by a weak van der Waals force, as shown in Figure 1c,d.

The shape of the as-synthesized PbI_2_ NPs was visualized by optical microscopy (OM). The OM confirmed the orientation distribution of triangular, pyramidal, and hexagonal PbI_2_ nanosheets synthesized using the simple solution method (Figure 2a–c). The SEM images further confirmed the uniformity and controllability of thickness in the three morphologies of the as-prepared PbI_2_ layered crystal (Figure 2d–h). The size and thickness of a single crystal of PbI_2_ estimated from OM and SEM are approximately 20 µm and 550–600 nm, respectively (Figure 2a–h). The results confirm that the solution-processed method effectively fabricates uniformly shaped 2D PbI_2_ layered crystals. The nucleation probability of PbI_2_ is determined using the following relation: A_1_ ∝ exp[−1/α^2^], where, A_1_ is the nucleation rate of crystals, and α is the supersaturation degree. The morphological change of the PbI_2_ crystal is derived from the difference between the growth rate of each crystal. As predicted by theoretical calculation, the low-index plane (001) of PbI_2_ has the lowest surface energy value [32,33]. The least surface energy value (0.48 J/m^2^) and low-index plane are favorable for nucleation probability. The growth rate of PbI_2_ NPs along the low-index (001) plane in the literal direction (*a*- and *b*-axis) is considerably higher than that in the vertical direction (*c*-axis), resulting in the formation of NPs with larger width-to-thickness ratio.

The crystal structure of as-grown 2D PbI_2_ NPs was characterized by XRD. As shown in Figure 3a, the four sharp and distinct diffraction peaks located at 12.24°, 25°, 38.19°, and 52.02° can be assigned to (001), (002), (003), and (004) crystal planes of hexagonal PbI_2_ (JCPDS, No.73-1750, space group: P-3m1), respectively [34]. These peaks indicate lamellar stacking along the *c*-axis of the I-Pb-I sandwich layer [35]. To enhance our comprehension of the physical characteristics of as-grown PbI_2_ nanoplates, we measured Raman spectroscopy using a 532 nm laser. This investigation yielded a spectrum featuring five distinct peaks within the 50–400 cm^−1^ range (Figure 3b). Notably, two prominent peaks at approximately 94 and 109 cm^−1^ correspond to the A_1g_ vibration mode associated with symmetrically stretched motion and the E_g_ vibration mode related to layered shear motion. Furthermore, we observed peaks at 165 and 213 cm^−1^, which are also present in mechanically exfoliated PbI_2_ single-crystal nanosheets. Notably, these latter two peaks become conspicuous only in thicker nanosheets [31]. Moreover, the optical properties of PbI_2_ NPs were analyzed. The UV–visible absorption spectrum in Figure 3c, shows an absorption peak centered at 498 nm. The optical bandgap estimated by Tauc plot corresponds to 2.36 eV (Figure 3d). The calculated bandgap is in good agreement with previous reports [36,37]. These results confirm the structural and optical quality of as-synthesized 2D PbI_2_ NPs, which are crucial for high-performance optoelectronic device fabrications.

The facile solution-processed method was employed for well-defined 2D PbI_2_ NPs that are suitable for optoelectronic applications. As a state-of-the-art application, a photodetector with ITO (150 nm)/PbI_2_/Au (100 nm) configuration was adopted to evaluate the optoelectronic properties of 2D PbI_2_ NPs. Comparatively, the Au top electrode (100 nm) was thermally evaporated (schematically illustrated in Figure 4a). Figure 4b shows the corresponding energy levels diagram of the photodetector; all the fundamental values are taken from the previous literature [36]. The transport mechanism of charge carriers in a self-powered photodetector can be comprehended using an energy level diagram (Figure 4b). Specifically, in the case of PbI_2_, the valence band maximum and conduction band minimum are situated at the approximate energy levels −6.02 and −4.1 eV, respectively. Consequently, when the photodetector is exposed to illumination, photogenerated carriers undergo spatial separation owing to the built-in potential. The work function of the ITO electrode is approximately −4.7 eV. As a result, photogenerated electrons are propelled towards the ITO side, while photogenerated holes migrate towards the Au electrode, driven by its work function of approximately −5.1 eV. During this process, the photogenerated holes must overcome an energy barrier of approximately 1 eV to complete the charge carrier cycle. The light response of the photodetector was studied under a 405 nm light illumination with different power densities (0.1, 0.3, 0.5, and 1.5 mW/cm^2^). An extremely low value of dark current (approximately 0.9 pA) and a photocurrent of 6.5 × 10^−8^ A at −2 V with an impressive photocurrent/dark current ratio of 10^4^ is observed from current vs. voltage (I-V) curves, which is good for high-performance narrow-band photodetectors (Figure 4c) [32]. We further extended our analysis by increasing the power density (0.1, 0.3, 0.5, and 1.5 mW/cm^2^). The results show an increase in photocurrent with the increasing power of the 405 nm illumination. This indicates that the efficiency of the photogenerated charge carriers is proportional to the number of photon flux absorbed.

Under visible-light illumination with a peak of 405 nm, the electron-hole pairs were generated, and the electrons were injected into the ITO side. Simultaneously, holes were collected through the Au electrode. The photodetector configuration used here is free of carrier transport layers, which would affect the operational stability of fabricated devices. Consequently, a better stability of the constructed device is expected. The rectifying characteristics are responsible for conferring the self-powered photodetector capability (Figure 4c), and this phenomenon can be attributed to the creation of Schottky junctions between the metal electrodes and the p-type PbI_2_ NPs. At an applied voltage of 0 V, the photocurrent exhibits a notable upward shift, providing further evidence of the self-powered nature of the device [18,38]. This phenomenon results from the energy difference between the anode and cathode, resulting in a built-in potential of approximately 0.11 V.

Analytical calculation was performed to evaluate photodetector performance. Photosensitivity (*K*) is the ability to distinguish an incident light from a dark condition [39,40]. It can be quantified as $K=\frac{I_{ill}-I_{dark\;}}{I_{dark}}$, where, *I_ill_* and *I_dark_* are photocurrent and dark current, respectively, and *K* = 3.9 × 10^3^ is achieved under 405 nm illumination at 0.1 mW/cm^2^. The photosensitivity *K* as a function of the power density is plotted in Figure 4d, showing an increasing trend with increasing power density. Additionally, key parameters such as photoresponsivity (*R*) and specific detectivity (*D**) are well-suited to assess photodetector performance, which can be defined as $R=\frac{I_{ph}}{P_{ill}}$, where, *P_ill_* is attributed to the power density. The maximum photoresponsivity is 0.51 mA/W at −2 V under 405 nm illumination at 0.1 mW/cm^2^. The specific detectivity (*D**) is the ability of a detector to distinguish the minimum impinging optical power and can be expressed as follows, $D^{*}=R\sqrt{\frac{S}{{2qI}_{dark}}}$, where, *q* is the elementary charge, *R* is the photoresponsivity, and *S* is the active area. *D** is calculated to be 2.5 × 10^12^ Jones at −2 V and is better than previously reported results (see Table 1).

Further, analytical calculations were performed for key performance parameters including open-circuit voltage (*V_oc_*), short-circuit current (*I_sc_*), on-off ratio, *R*, and *D** at zero applied bias, whereas the photodetector operates in self-powered mode. The calculated values are as follows: *V_oc_* = 0.103 V, *I_sc_* = 1.93 × 10^−8^, on-off ratio = 10^3^, *R* = 4.0 mA/W, and *D** = 3.3 × 10^11^ Jones. The *R* and *D** (−2 V) as a function of power densities are plotted in Figure 4d,e. *R* and *D** decrease as a function of increasing power density owing to the compound loss in the device. All key values are comparable to CVD/PVD-grown PbI_2_ NPs (see Table 1).

The photocurrent response indicates the durability of optoelectronic devices for practical applications. The transient photo-response of our photodetector was measured by periodically switching a 405 nm light with a light intensity of 5 mW/cm^2^ at −5 V at a constant interval for several cycles (Figure 5a). When the light is turned on, the photocurrent is generated, whereas it quickly decays when the light is turned off, as shown in Figure 5a. The highly stable and repeatable photo-switching performance of the 2D PbI_2_ NPs photodetector can be attributed to the fact that the photo-response remains almost unchanged even after 250 s of continuous operation. The rise and decay times of the photodetector are the times taken for the photocurrent to increase from 10% to 90% of the peak value and vice versa. As plotted in Figure 5b,c, the rise and decay time of our device is 0.23 and 0.38 s, respectively. The slow response time observed in our device can be attributed to a substantial density of surface defects, which originated from the employed solution-processible method. The phenomenon of photogeneration primarily takes place within the charge traps or surface adsorbates, causing one type of charge carrier to exit while the other is entrapped within these trap states. Such behavior usually occurs in 2D layered materials, which possess large surface-to-volume ratios [41,42]. Therefore, the results were believed to provide a deep insight into the efficient application of PbI_2_ nanosheets for high-performance optoelectronic devices.

To further illustrate the potential feasibility of the self-powered capability of the device, we conducted temporal response measurements, subsequent to cyclic light modulation, at an illumination intensity of 27.4 mW/cm^2^. The resulting fabricated device exhibited exceptional reproducibility and stability over various time intervals (200 s), manifesting rise and decay values lower than 1 s (Figure 5d). This self-powered performance can be ascribed to the inherent potential gradient originating from the energy difference between the anode and cathode.
micromachines-14-01949-t001_Table 1Table 1Comparison of various key properties of different representative PbI_2_-based photodetectors.Device ConfigurationGrowth MechanismLight IntensityPhotosensitivity(*K*)Responsivity(*R*)Detectivity*D** (Jones)Rise/Fall TimeRef.ITO/PbI_2_/AuSolution-process0.1 mW/cm^2^3.9 × 10^3^0.5 mA/W2.5 × 10^12^0.23/0.38 s (5 mW/cm^2^)This WorkITO/PbI_2_/NiSolution-process1.14 mW/mm^2^-0.65 A/W0.95 × 10^13^2/3 ms[36]SiO_2_/Si/PbI_2_/AuPVD-grown ^†^40 mW/cm^2^---18/22 ms[32]PET/Graphene/PbI_2_/GraphenePVD-grown5 µW/cm^2^-45 A/W-35/20 µs[30]SiO_2_/Si/SbSI/PbI_2_/AgHydrothermal method0.1 mW/cm^2^-26.3 mA/W-12/8 ms[43]Ti/Au/PbI_2_/AuSolution-process0.17 mW/cm^2^-40 mA/W3.31 × 10^10^161.7/192 ms[33]Si/PbI_2_-MAPbI_2_/Ti/AuPVD-grown--410 mA/V3.1 × 10^11^1.4/0.9 s[44]Au/PbI_2_/AuPVD-grown3.4 mW/cm^2^-147.6 A/W2.56 × 10^12^18/25 ms[35]SiO_2_/Si/WS_2_/PbI_2_/AuPVD-grown0.01 mW/cm^2^-7.1 × 10^4^ A/W-26.4/28.9 ms[45]Polyimide/PbI_2_/AuHydrothermal method--5 mA/W-30 ms[46]Si/SiO_2_/PbI_2_/AuPVD-process--13 mA/W-425/41 ms[47]MoO_3_/Iridium/SiAtomic layer deposition--34 A/W7 × 10^11^0.1 ms[48]*β*-Ga_2_O_3_/TAPCMetal–organic chemical vapor deposition100 mW/cm^2^-1.41 mA/W1.02 × 10^13^-[49]Mo/Sb_2_Se_3_/CdS(Al)/ITO/AgThermodynamic/kinetic deposition11 nW/cm^2^-0.9 A/W4.78 × 10^12^24/75 ns[50]^†^ PVD-physical vapor deposition.

## 4. Conclusions

In summary, solution-processed photodetectors based on 2D PbI_2_ NPs were fabricated. The structural and optical properties of the as-synthesized PbI_2_ nanoplates were carefully analyzed and the nucleation and growth mechanism was briefly explained based on structural properties. The PbI_2_ nanosheets synthesized were used to fabricate photodetectors with the asymmetric electrode configuration ITO/PbI_2_/Au. The photodetector showed a light response at 0 V bias under 405 nm light illumination with a light intensity of 0.1 mW/cm^2^, which was attributed to the variation in the work function of the electrodes. The device configuration (ITO/PbI_2_ NPs/Au) achieved high photoresponsivity and specific detectivity of 0.51 mA/W and 2.5 × 10^12^ Jones at −2 V, respectively. Further, analytical calculations were performed for key performance parameters including open-circuit voltage (*V_oc_*), short-circuit current (*I_sc_*), on-off ratio, *R*, and *D** at zero applied bias, whereas the photodetector was operating in self-powered mode. The calculated values are as follows: *V_oc_* = 0.103 V, *I_sc_* = 1.93 × 10^−8^, on-off ratio = 10^3^, *R* = 4.0 mA/W, and *D** = 3.3 × 10^11^ Jones. Additionally, the device exhibited a quick response of 0.23/0.38 s at −5 V and approximately 1 s at 0 V under visible light. Based on the results presented here with PbI_2_ NPs and their promising application to photodetectors, future advances in processing and fabrication techniques are anticipated.

## Figures and Tables

**Figure 1 micromachines-14-01949-f001:**
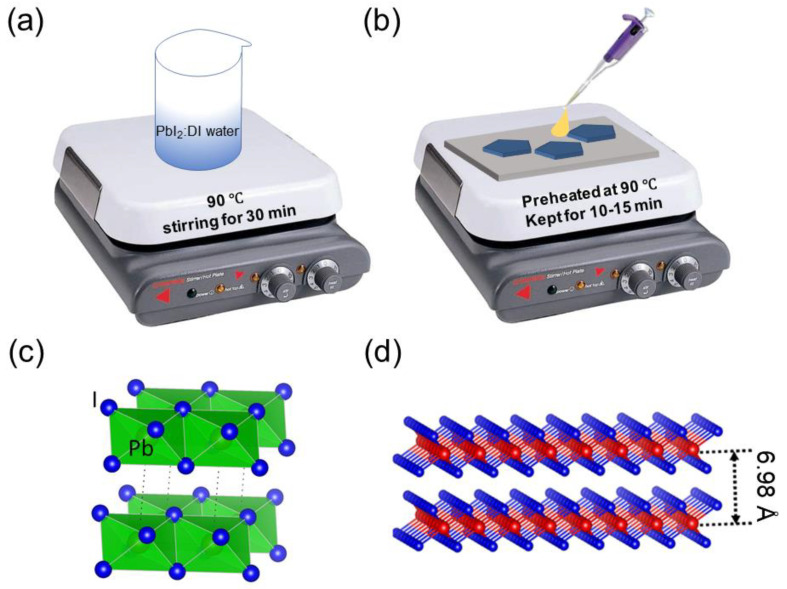
(**a**) Schematic of 2D PbI_2_ precursor preparation; (**b**) the 2D PbI_2_ NP deposition on a glass substrate; (**c**) single unit cell of PbI_2_; (**d**) layered PbI_2_ crystal structure separated by a distance of 6.98 Å.

**Figure 2 micromachines-14-01949-f002:**
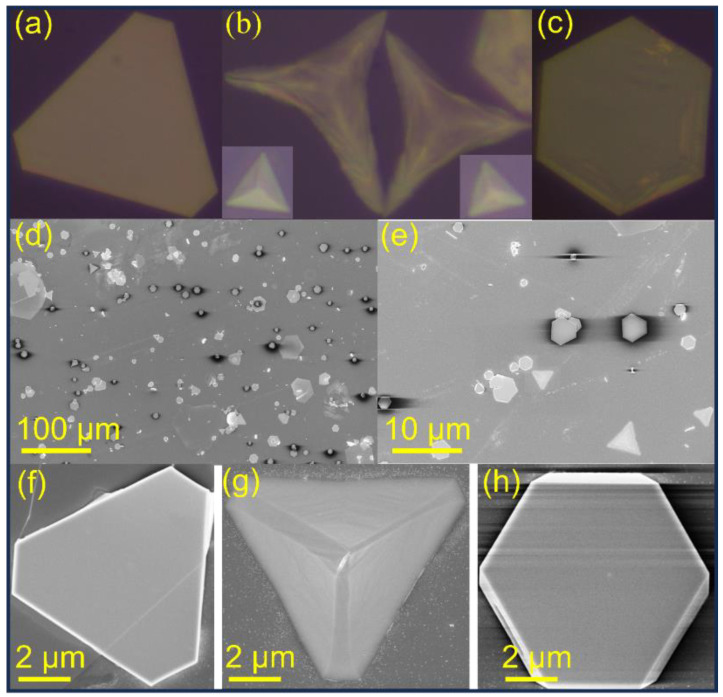
(**a**–**c**) Optical microscopic images of PbI_2_ NPs with different morphologies. (**d**,**e**) SEM images of PbI_2_ NPs. (**f**–**h**) SEM images of PbI_2_ NPs with different structural morphologies.

**Figure 3 micromachines-14-01949-f003:**
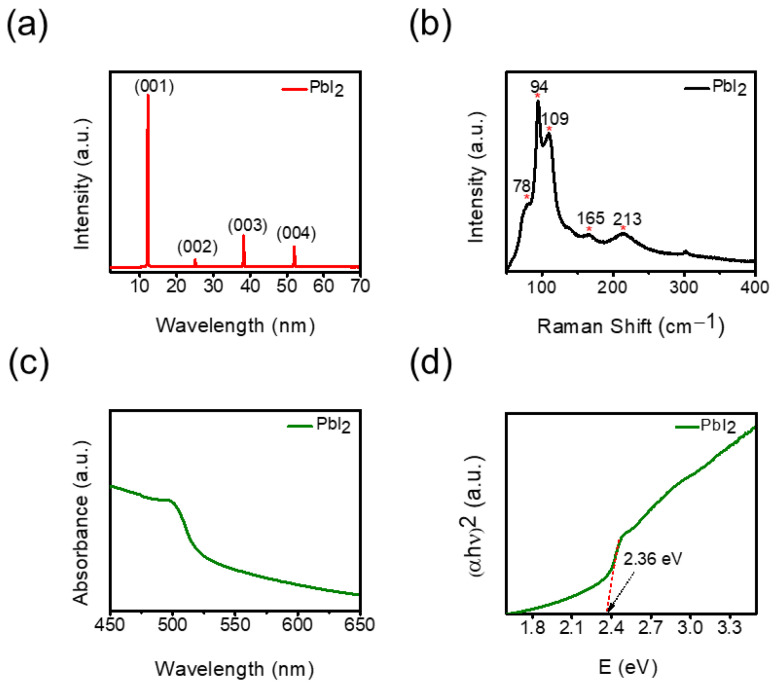
(**a**) XRD pattern of PbI_2_ NPs. (**b**) Raman spectrum of PbI_2_ nanosheets measured with a 532 nm laser light. (**c**) The UV absorption spectrum of PbI_2_ NPs. (**d**) Tauc plot of as-grown two-dimensional PbI_2_ NPs.

**Figure 4 micromachines-14-01949-f004:**
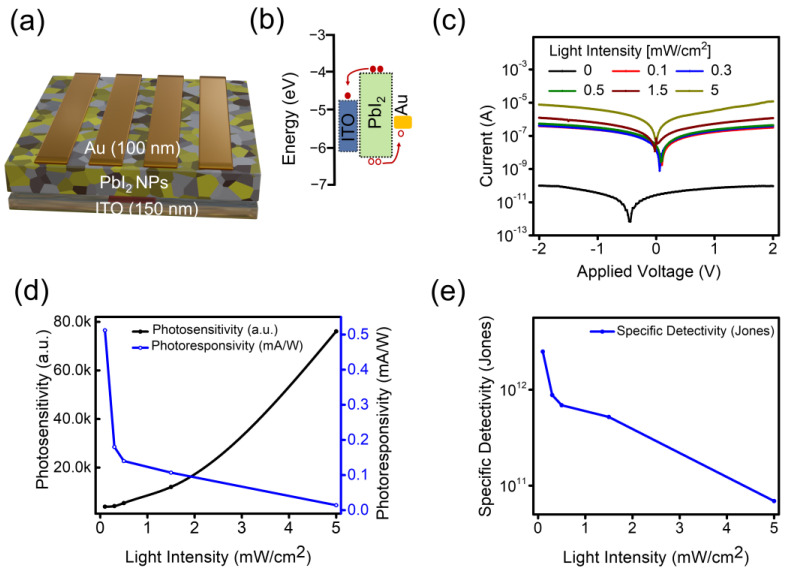
(**a**) Cross-section image of PbI_2_ NP photodetector. (**b**) Band diagram illustrating the charge transport mechanism of the photodetector at zero bias. (**c**) Current-voltage curve of a photodetector in the dark and under the illumination of light with different intensities. (**d**) Photosensitivity and photoresponsivity as a function of light intensity. (**e**) Specific detectivity (*D**) as a function of light intensity.

**Figure 5 micromachines-14-01949-f005:**
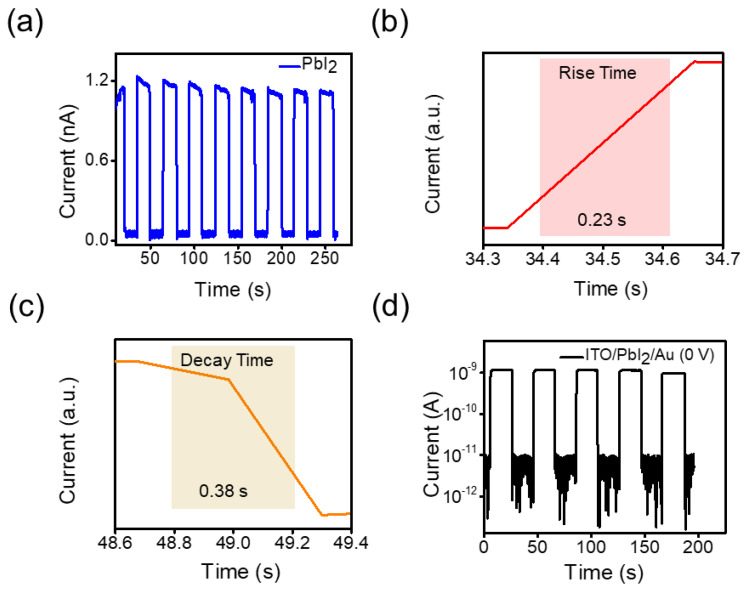
(**a**) Temporal response of the PbI_2_ photodetector. (**b**) Rising time, (**c**) decay time, and (**d**) temporal response of the self-powered photodetector.

## Data Availability

All the relevant data are given in this paper.

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
