# Peer review of "Aqueous-Phase Formation of Two-Dimensional PbI2 Nanoplates for High-Performance Self-Powered Photodetectors"

_micromachines, 2023, doi:10.3390/mi14101949_

Round 1
Reviewer 1 Report
Aqueous-phase formation of two-dimensional PbI2 nanoplates for high-performance self-powered photodetector
The manuscript entitled “Aqueous-phase formation of two-dimensional PbI2 nanoplates for high-performance self-powered photodetector” is presented by M.I. Saleem, et al. The manuscript is well-composed and shows promise, but it needs substantial revisions to improve its scientific merit and ease of reading. Addressing the highlighted issues will greatly enhance the study's clarity and impact.
Recommendation: Major revision
1- The author asserts that PbI2 operates as a self-powered device. Can the current-versus-time response at zero applied bias be demonstrated?
2- Can the drop-casting technique be employed to fabricate the large-area photodetector while managing the complexities associated with this method?
3- I'm not entirely clear on the growth mechanics of PbI2. Could the author please provide a clear demonstration of how PbI2 nanoplates were synthesized?
4- To substantiate your claim that the photodetector is self-powered, it is recommended to compute the responsivity and detectivity values when the photodetector is operating under zero-bias conditions. Additionally, please include the values for the short-circuit current (ISC) and open-circuit voltage (VOC) to provide a more comprehensive assessment of the photodetector's self-powered capabilities.
5- In Figure 5(b), the author reports a rise time of 0.23 seconds, yet the text cites a rise time of 0.21 seconds. Could you please clarify why there is an inconsistency between these two values? Additionally, in Table 1, the rise and fall times are calculated at an illumination intensity of 0.1 mW/cm2, which is different from what is mentioned in the text. Could you please explain the discrepancy?
6- The presence of a non-zero dark current is noted. Could you please provide an explanation for the underlying mechanisms contributing to this phenomenon? Additionally, it would be valuable to address the issues of repeatability and stability in the context of this reference.
7- The device physics and carrier transportation mechanisms for photogenerated electrons need to be discussed with the help of an energy band diagram. Some literature is suggested to mention for device physics 10.1021/acsami.3c01749; 10.1016/j.sna.2019.07.003; 10.3390/nano12172962; 10.1016/j.sna.2019.01.029; 10.1002/admi.202200105; 10.1016/j.sna.2021.113073; 10.1016/j.surfin.2022.101772, 10.1002/admi.202200017; 10.1016/j.mtphys.2022.100829; 10.1002/adfm.202201527; 10.1007/s10825-020-01583-6; 10.1021/acsami.8b00058; 10.1002/adma.202002628; 10.1016/j.jphotochem.2021.113764; 10.1021/acsami.0c04093; 10.1002/admi.202000360.
Extensive editing of English language required
Author Response
Reviewer #1:
- The author asserts that PbI2operates as a self-powered device. Can the current-versus-time response at zero applied bias be demonstrated?
Reply: We thanks for the Reviewer’s insightful feedback.
Addition:
(Page 8, line 267-273 in the revised manuscript) “To further illustrate the potential feasibility of the self-powered capability of the device, we conducted temporal response measurements, subsequent to cyclic light modulation, at an illumination intensity of 27.4 mW/cm2. The resulting fabricated device exhibited exceptional reproducibility and stability over various time intervals (200 s), manifesting rise and decay values lower than 1 s (Fig. 5d). This self-powered performance can be ascribed to the inherent potential gradient originating from the energy difference between the anode and cathode.
- Can the drop-casting technique be employed to fabricate the large-area photodetector while managing the complexities associated with this method?
Reply: We express our gratitude for the valuable insights provided by the reviewer. The lateral dimensions and thickness of the PbI2 nanosheets, as illustrated in Figure 2a-h, measure approximately 20 µm and 550–600 nm, respectively. We hold a strong belief in the considerable potential of PbI2 for application in large-area devices. Nevertheless, it is important to note that the exploration of such utilization is currently under contemplation.
- I'm not entirely clear on the growth mechanics of PbI2. Could the author please provide a clear demonstration of how PbI2nanoplates were synthesized
Reply: Sorry and thank you for your comment. We have revised the growth mechanics of PbI2 nanosheets.
Modification:
(Page 3, line 105-110 in the revised manuscript) “First, PbI2 (10 mg) was mixed in 10 mL of deionized water and magnetically stirred at 90°C until the powder dissolved entirely. The solution became clear with negligible sedimentation. Subsequently, 50–60 µL of the clear supernatant solution was dropped onto an oxygen-plasma glass substrate in ambient condition at 90°C. Subsequently, the temperature was elevated to 180°C within 5 min to support nucleation and growth with distinct geometric configurations, including triangular, pyramidal, and hexagonal PbI2 nanosheets”.
- To substantiate your claim that the photodetector is self-powered, it is recommended to compute the responsivity and detectivity values when the photodetector is operating under zero-bias conditions. Additionally, please include the values for the short-circuit current (ISC) and open-circuit voltage (VOC) to provide a more comprehensive assessment of the photodetector's self-powered capabilities.
Reply: We thanks for the Reviewer’s insightful feedback.
Addition:
(Page 7, from line 238 to line 242 in the revised manuscript)
“Further, analytical calculations have been performed for key performance parameters including open-circuit voltage (Voc), short-circuit current (Isc), on-off ratio, R, and D* at zero applied bias, whereas the photodetector is operating in self-powered mode. The calculated values are as follows: Voc =0.103 V, Isc =1.93´10-8, on-off ratio =103, R =4.0 mA/W, and D* =3.3 ×1011 Jones.”.
- In Figure 5(b), the author reports a rise time of 0.23 seconds, yet the text cites a rise time of 0.21 seconds. Could you please clarify why there is an inconsistency between these two values? Additionally, in Table 1, the rise and fall times are calculated at an illumination intensity of 0.1 mW/cm2, which is different from what is mentioned in the text. Could you please explain the discrepancy?
Reply: Thank you for the comments and sorry for the mistake. We have corrected the rise time value in the revised manuscript. Further, we have calculated the temporal response under the illumination of 5 mW/cm2. Thus, a revision has been made in Table 1.
- The presence of a non-zero dark current is noted. Could you please provide an explanation for the underlying mechanisms contributing to this phenomenon? Additionally, it would be valuable to address the issues of repeatability and stability in the context of this reference.
Reply: We express our gratitude for the valuable insights provided by the Reviewer in their feedback. This is normally due to a large leakage current, which causes non-zero dark current in fabricated photodetector.
- The device physics and carrier transportation mechanisms for photogenerated electrons need to be discussed with the help of an energy band diagram. Some literature is suggested to mention for device physics 10.1021/acsami.3c01749; 10.1016/j.sna.2019.07.003; 10.3390/nano12172962; 10.1016/j.sna.2019.01.029; 10.1002/admi.202200105; 10.1016/j.sna.2021.113073; 10.1016/j.surfin.2022.101772, 10.1002/admi.202200017; 10.1016/j.mtphys.2022.100829; 10.1002/adfm.202201527; 10.1007/s10825-020-01583-6; 10.1021/acsami.8b00058; 10.1002/adma.202002628; 10.1016/j.jphotochem.2021.113764; 10.1021/acsami.0c04093; 10.1002/admi.202000360.
Reply: We express our gratitude for the valuable insights provided by the reviewers in their feedback.
Revision: We have conducted a comprehensive examination of the device physics and carrier transport mechanisms pertaining to photogenerated electrons and holes, utilizing energy band diagrams as a valuable tool for elucidation. All literature has been cited, as suggested by reviewers.
Addition
(Page 5, from line 184 to line 193 in the revised manuscript)
“The transport mechanism of charge carriers in a self-powered photodetector can be comprehended using an energy level diagram (Fig. 4b). Specifically, in the case of PbI2, the valence band maximum and conduction band minimum are situated at the approximate energy levels -6.02 and -4.1 eV, respectively. Consequently, when the photodetector is exposed to illumination, photogenerated carriers undergo spatial separation owing to the built-in potential. The work function of the ITO electrode is approximately -4.7 eV. As a result, photogenerated electrons are propelled towards the ITO side, while photogenerated holes migrate towards the Au electrode, driven by its work function of approximately -5.1 eV. During this process, the photogenerated holes must overcome an energy barrier of approximately 1 eV to complete the charge carrier cycle”.
Reviewer 2 Report
The wok is very fine and interesting. I suggest to accept this manuscript with following two miner notes.
1. Could authors provide Raman spectra of the fabricated 2D plates? Since Raman spectra is strong and very effective tool to declare the crystal structure of the material.
2. Could the author comment on the controllability of the shape and size of the PbI2 plates?
Author Response
Reviewer #2:
Reviewer#2: The work is very fine and interesting. I suggest to accept this manuscript with following two miner notes.
We appreciate your effort in reviewing our work and the constructive criticism that contributes to enhancing the quality of the manuscript. Considering the comments raised, we have worked diligently to offer a thorough response to the revised manuscript.
- Could authors provide Raman spectra of the fabricated 2D plates? Since Raman spectra is strong and very effective tool to declare the crystal structure of the material.
Reply: We express our gratitude to the reviewer for providing valuable feedback. We have measured Raman Spectrum of PbI2 nanosheets and data is presented in the revised manuscript
Addition:
(Page 4, from line 161 to line 169 in the revised manuscript)
“To enhance our comprehension of the physical characteristics of as-grown PbI2 nanoplates, we measured Raman spectroscopy using a 532 nm laser. This investigation yielded a spectrum featuring five distinct peaks within the 50-400 cm-1 range (Figure 3b). Notably, two prominent peaks at approximately 94 and 109 cm-1 correspond to the A1g vibration mode associated with symmetrically stretched motion and the Eg vibration mode related to layered shear motion. Furthermore, we observed peaks at 165 and 213 cm-1, which are also present in mechanically exfoliated PbI2 single-crystal nanosheets. Notably, these latter two peaks become conspicuous only in thicker nanosheets”.
- Could the author comment on the controllability of the shape and size of the PbI2 plates?
Reply: We express our gratitude to the reviewer for providing valuable feedback aimed at enhancing the quality of the manuscript. Despite our efforts, we encountered challenges in precisely controlling the size and shape of PbI2 nanoplates, as evident from the Optical microscopy and SEM images.
Reviewer 3 Report
· How is the shape of the Au electrodes on the surface of your thin film? Does it matter to use only Au?
· Compared with some recent self-powered photodetectors, such as
Self‐Powered UV Photodetector Utilizing Plasmonic Hot Carriers in 2D α‐MoO3/Ir/Si Schottky Heterojunction Devices - Basyooni - physica status solidi (RRL) – Rapid Research Letters - Wiley Online Library
Wu, C., Wu, F., Ma, C., Li, S., Liu, A., Yang, X., ... & Guo, D. (2022). A general strategy to ultrasensitive Ga2O3 based self-powered solar-blind photodetectors. Materials Today Physics, 23, 100643.
Chen, S., Fu, Y., Ishaq, M., Li, C., Ren, D., Su, Z., ... & Tang, J. (2023). Carrier recombination suppression and transport enhancement enable high‐performance self‐powered broadband Sb2Se3 photodetectors. InfoMat, e12400.
· It is interesting that you use Au and still have Schotaky contact effects. how come?
· The introduction part should be enhanced with a literature review more about the self-powered photodetectors, mechanisms behind
· Your response time is a bit long compared to the huge number of photodetectors, can you mention why?
· Raman data should provide and mention how many layers your 2D materials are.
· Fig 4 a, why detetctivivty decreases with light intensity,? It should be increased right? Is an internal heating source the reason?
· Remove sec from Table 1 of the response time section
· How can you enhance the response of your sensors? Can you suggest some potential ideas?
Moderate editing of English language required
Author Response
Reviewer #3:
- How is the shape of the Au electrodes on the surface of your thin film? Does it matter to use only Au? Compared with some recent self-powered photodetectors, such as
Reply: We express our gratitude for the valuable insights provided by the reviewers in their feedback.
Revision: We assessed the performance of PbI2 nanoplates by substituting the Au electrode with an Ag electrode. The photodetector is functioning as expected, and we are presenting the results of ITO/PbI2/Ag for your review.
(Addition)
The performance of the recommended articles (I, II, and III) has been assessed, and all the key parameters have been compared and listed in Table 1.
- Self‐Powered UV Photodetector Utilizing Plasmonic Hot Carriers in 2D α‐MoO3/Ir/Si Schottky Heterojunction Devices - Basyooni - physica status solidi (RRL) – Rapid Research Letters - Wiley Online Library
- Wu, C., Wu, F., Ma, C., Li, S., Liu, A., Yang, X., ... & Guo, D. (2022). A general strategy to ultrasensitive Ga2O3 based self-powered solar-blind photodetectors. Materials Today Physics, 23, 100643.
- Chen, S., Fu, Y., Ishaq, M., Li, C., Ren, D., Su, Z., ... & Tang, J. (2023). Carrier recombination suppression and transport enhancement enable high‐performance self‐powered broadband Sb2Se3 InfoMat, e12400.
- It is interesting that you use Au and still have Schottky contact effects. how come?
Reply: Thank you for your comments. The junction between metal and semiconductor materials is commonly referred to as a Schottky junction. The formation of this junction is not dependent on the specific metal electrode used. For instance, as documented in the following literature [1-4].
3. The introduction part should be enhanced with a literature review more about the self-powered photodetectors, and the mechanisms behind them.
Reply: We express our gratitude for the valuable insights provided by the Reviewers in their feedback.
Addition
(Page 2, from line 70 to line 86 in the revised manuscript)
“Recently, photodetectors are gaining interest, owing to their fundamental capability of converting incident light pulses into electrical signals [14-19]. Photodetectors find extensive applications in various fields such as infrared imaging, fire imaging, optical communications, biological sensing, and spectroscopy [20-29]. The remarkable photophysical characteristics and structural advantages exhibited by 2D layered materials have positioned them at the forefront of research for photodetection applications.
Graphene has been a prominent choice in photodetection applications; however, the lack of a discernible bandgap restricts its applicability. Consequently, other 2D layered materials, including black phosphorus (BP), have emerged as promising alternatives. Nevertheless, the use of BP in photodetection is limited by stability concerns, particularly its susceptibility to irreversible oxidation when exposed to ambient conditions.
The pursuit of autonomously operating photodetectors, leveraging stable 2D layered materials within their design, holds considerable promise. They have the unique capability of harnessing energy from their surrounding environment and converting it into electrical pulses. This is an essential characteristic for practical applications owing to several key advantages, including low energy consumption, sustainability, and simplified design of driving circuits”.
4. Your response time is a bit long compared to the huge number of photodetectors, can you mention why?
Reply: Thank you for your comments. We have reasoned this phenomenon owing to surface traps or surface adsorbates.
Addition
(Page 8, from line 259 to line 264 in the revised manuscript)
“The slow response time observed in our device can be attributed to a substantial density of surface defects, which originated from the employed solution-processible method. The phenomenon of photogeneration primarily takes place within the charge traps or surface adsorbates, causing one type of charge carriers to exit while the other are entrapped within these trap states. Such behavior usually occurs in 2D layered materials, which possess large surface-to-volume ratios [41, 42]”.
5. Raman data should provide and mention how many layers your 2D materials are.
Reply: We express our gratitude to the reviewer for providing valuable feedback.
Addition:
(Page 4, from line 161 to line 169 in the revised manuscript)
“To enhance our comprehension of the physical characteristics of as-grown PbI2 nanoplates, we measured Raman spectroscopy using a 532 nm laser. This investigation yielded a spectrum featuring five distinct peaks within the 50-400 cm-1 range (Figure 3b). Notably, two prominent peaks at approximately 94 and 109 cm-1 correspond to the A1g vibration mode associated with symmetrically stretched motion and the Eg vibration mode related to layered shear motion. Furthermore, we observed peaks at 165 and 213 cm-1, which are also present in mechanically exfoliated PbI2 single-crystal nanosheets. Notably, these latter two peaks become conspicuous only in thicker nanosheets”.
6. Fig 4 a, why detectivity decrease with light intensity,? It should be increased right? Is an internal heating source the reason?
Reply: We express our gratitude to the reviewer for their valuable comments. The photoresponsivity decreases with increasing light intensity It can be explained as follows: when light irradiates on semiconducting, three different processes namely, electron-hole pair generation, electron-hole pair recombination, and motion of generated carriers under strong built-in electric field take place [5-7]. Photogenerated carriers are proportional to light intensity and its absorption into the material. At low light intensity, a strong built-in electric field reduces the rate of recombination, resulting in high spectral responsivity. However, at higher light intensities, the increased rate of recombination because of reduced built-in potential lowers the responsivity [6-8]. It is important to note that the specific detectivity is directly linked to the photoresponsivity; thus, it also diminishes as light intensity increases.
7. Remove sec from Table 1 of the response time section
Reply: Thank you for your comments.
Modification:
We have removed the word “sec” from Table 1.
8. How can you enhance the response of your sensors? Can you suggest some potential ideas?
Reply: Thank you for your comments. We are exploring the possibility of fabrication type-II heterojunction to improve the performance of all-solution processed self-powered devices, including those based on MoX2 (S, Se, Te)/PbI2, WS2/PbI2, and graphene/PbI2. This work is still in the contemplation phase.
Round 2
Reviewer 1 Report
no further revision is required.
Reviewer 3 Report
you must check Table 1 , I think there is a problem in the formatting
Minor editing of English language required